High-risk histological subtype-related FAM83A hijacked FOXM1 transcriptional regulation to promote malignant progression in lung adenocarcinoma

Fei Wei 1
Yan Yan 2
Liu Guangjun 3
Peng Bo 1 3
Liu Yuanyuan 4 hbulyy@126.com
Chen Qiang 1 3 candycy1119@163.com
1 Department of Clinical College, Xuzhou Medical University , Xuzhou, Jiangsu , China
2 Department of Cardiovascular Medicine, The Affiliated Hospital of Xuzhou Medical University , Xuzhou, Jiangsu , China
3 Department of Thoracic Surgery, Xuzhou Central Hospital , Xuzhou, Jiangsu , China
4 Department of Respiratory and Critical Care Medicine, Xuzhou Central Hospital , Xuzhou, Jiangsu , China
Tyagi Abhishek
Electronic publication date: 2023 Oct 26
Publication date: 2023
Volume: 11
Electronic Location ID: e16306
Received 2023 May 26; Accepted 2023 Sep 26
Copyright: © 2023 Fei et al.
Copyright year: 2023
Copyright holder: Fei et al.
License: This is an open access article distributed under the terms of the Creative Commons Attribution License, which permits unrestricted use, distribution, reproduction and adaptation in any medium and for any purpose provided that it is properly attributed. For attribution, the original author(s), title, publication source (PeerJ) and either DOI or URL of the article must be cited.
License URL: https://creativecommons.org/licenses/by/4.0/

Keywords: LUAD, FAM83A, Histological pathology subtype, Transcription factors

Funding: General Program of Xuzhou Science and Technology Bureau project KC20103 Affiliated Hospital of Xuzhou Medical University XYFM2020002 This work was supported by the General Program of Xuzhou Science and Technology Bureau project [KC20103], and the Scientific and Technological Development Project of the Affiliated Hospital of Xuzhou Medical University [XYFM2020002]. The funders had no role in study design, data collection and analysis, decision to publish, or preparation of the manuscript.

==============================
Background

According to the histopathology, lung adenocarcinoma (LUAD) could be divided into five distinct pathological subtypes, categorized as high-risk (micropapillary and solid) group, intermediate-risk (acinar and papillary) group, and low-risk (lepidic) group. Despite this classification, there is limited knowledge regarding the role of transcription factors (TFs) in the molecular regulation of LUAD histology patterns.

Methods

Publish data was mined to explore the candidate TFs associated with high-risk histopathology in LUAD, which was validated in tissue samples. Colony formation, CCK8, EdU, transwell, and matrigel assays were performed to determine the biological function of FAM83A in vitro. Subcutaneous tumor-bearing in BALB/c nude mice and xenograft perivitelline injection in zebrafish were utilized to unreal the function of FAM83A in vivo. We also performed chromatin immunoprecipitation (ChIP), dual-luciferase reporter, and rescue assays to uncover the underline mechanism of FAM83A. Immunohistochemistry (IHC) was performed to confirm the oncogenic role of FAM83A in clinical LUAD tissues.

Results

Screening the transcriptional expression data from TCGA-LUAD, we focus on the differentially expressed TFs across the divergent pathological subtypes, and identified that the expression of FAM83A is higher in patients with high-risk groups compared with those with intermediate or low-risk groups. The FAM83A expression is positively correlated with worse overall survival, progression-free survival, and advanced stages. Gain- and loss-of-function assays revealed that FAM83A promoted cell proliferation, invasion, and migration of tumor cell lines both in vivo and in vitro. Pathway enrichment analysis shows that FAM83A expression is significantly enriched in cell cycle-related pathways. The ChIP and luciferase reporter assays revealed that FAM83A hijacks the promoter of FOXM1 to progress the malignant LUAD, and the rescue assay uncovered that the function of FAM83A is partly dependent on FOXM1 regulation. Additionally, patients with high FAM83A expression positively correlated with higher IHC scores of Ki-67 and FOXM1, and patients with active FAM83A/FOXM1 axis had poor prognoses in LUAD.

Conclusions

Taken together, our study revealed that the high-risk histological subtype-related FAM83A hijacks FOXM1 transcriptional regulation to promote malignant progression in lung adenocarcinoma, which implies targeting FAM83A/FOXM1 is the therapeutic vulnerability.

Introduction

Lung cancer is the leading cause of cancer-related deaths worldwide, which accounts for more than 20% of all cancer deaths in 2020 (Siegel, Miller & Jemal, 2020). Non-small cell lung cancer (NSCLC) is the most common type of lung cancer, representing approximately 85% of all cases. Lung adenocarcinoma (LUAD) is a prominent histologic subtype, accounting for 47% of all NSCLC cases (Gridelli et al., 2015). Intratumoral heterogeneity is a hallmark of LUAD, significantly impacting the effectiveness of treatments and not yet fully understood. The classification of LUAD into five major histologic subtypes based on morphological characteristics, namely lepidic (LEP), acinar (ACI), papillary (PAP), micropapillary (MIP), and solid (SOL), has helped to resolve this spatial heterogeneity (SOL) (Travis et al., 2011). According to the different prognosis for disease-free survival and recurrence, they are classified into three groups: low-risk group (LEP), intermediate-risk group (ACI and PAP), and high-risk group (MIP and SOL) (Sica et al., 2010; Ujiie et al., 2015).

Researchers have made efforts to address the underlying mechanism to determine the morphological features and various prognosis across the major subtypes. In a recent study by Caso et al. (2020), targeted next-generation sequencing was employed to analyze the genomic features of tumors obtained from 604 patients with stage I to III LUAD who underwent complete resection. Results indicated that MIP/SOL tumors exhibited higher levels of tumor mutational burden, whole-genome doubling rates and alterations in a greater number of oncogenic pathways compared to LEP and ACI/PAP tumors. Sato et al. (2021) uncovered that histological characteristics depend on the site of development, and a solid-to-acinar transition (SAT) could be induced by the tumor microenvironment. Their findings demonstrated that cancer-associated fibroblasts (CAFs) activate TGF-β signaling to remodel tumor tissues, ultimately determining the histological pattern of LUAD. Nguyen et al. (2022) observed distinct gene expression patterns between the lepidic and solid subtypes, while the remaining subtypes showed similarity in terms of gene expression. To facilitate this observation, they developed two gene signatures, referred to as L-score and S-score, respectively. Patients with higher L-score exhibited prolonged survival and a higher likelihood of responding positively to immune checkpoint blockade therapy. However, litter is known about how transcription factors (TFs) determine the morphological features and malignant progression of histological subtypes.

Transcription factors have a DNA-binding domain that allows them to recognize DNA sequences, and recruit the mediator complex or co-factors for transcription initiation (Badis et al., 2009). Dysregulated transcription is the hallmark of cancer, and the abnormally expressed transcription factors play important roles in perturbing transcription and cell fate determination. Lenaerts et al. (2022) revealed that the B-lineage determinant transcription factor EBF1 regulates lineage preference in early progenitors. EBF1 constrained C/EBPα-driven myelopoiesis and primed the B-lymphoid fate, which regulated myeloid/lymphoid fate bias. During embryonic development and in pathological conditions, the transcription factor SOX9 is activated at the onset of endothelial-to-mesenchymal transition, which displays features of a pioneer transcription factor, such as opening of chromatin and leading to deposition of active histone modifications at silent chromatin regions (Fuglerud et al., 2022). We hypothesized that the transcription factors have critical roles in morphological features determination and malignant progression of histological subtypes in LUAD.

In this study, we screened for differentially expressed transcription factors across the three subtype groups and identified that the expression of FAM83A was significantly higher in the high-risk group in comparison to the intermediate- or low-risk group. This positively correlation between FAM83A expression and worse overall and progression-free survival, as well as advanced stages, was observed. Gain- and loss-of-function assays revealed that FAM83A promoted cell proliferation, invasion, and migration of tumors both in vitro and in vivo. Further mechanistic studies revealed that FAM83A hijacked the promoter region of the FOXM1 to regulate the cell cycle and malignant progression. Furthermore, patients with an active FAM83A/FOXM1 axis were found to have poor prognosis in LUAD, which implied targeting FAM83A/FOXM1 may represent a therapeutic vulnerability for LUAD patients with high-risk histopathology.

Materials and Methods

Patients and tissue samples

In order to conduct additional research, LUAD and nearby normal tissues were taken from 12 patients at the Xuzhou Central Hospital and preserved at −80 °C. Pathologists from our facility verified all cancers and the corresponding normal tissues. None of the participants in this study had any other cancers, nor had they undergone any radiation or chemotherapy prior to surgery. Each patient’s clinical and pathological characteristics were subsequently gathered following their surgical procedure. All patients gave written informed permission for the study, and it complied with Xuzhou Central Hospital Ethics Committee standards and the Ethics Committee of Jiangsu Cancer Hospital. The Ethics Committee of Jiangsu Cancer Hospital approved the study.

Cell culture

Human LUAD cell lines (A549, PC9, H1975, A427, H358 and DV90) and normal lung epithelial cells (HBE) were purchased from the Chinese Academy of Sciences (Shanghai, China). The aforementioned cells were grown in DMEM media (Invitrogen, Carlsbad, CA, USA), 10% FBS, and 1% penicillin/streptomycin supplements, all from the same source, at 37 °C and 5% CO2. The media was then replaced every 2 to 3 days while cell growth was periodically monitored. For the following studies, cells in the logarithmic growth phase were chosen. According to earlier investigations (Goel et al., 2017), palbociclib (HY-50767, MCE) was added to the cell media at a concentration of 1 nM, incubated for 48-h.

Cell transfection

Target FAM83A’s sgRNA and FOXM1’s siRNA were designed by GenScript GenCRISPR gRNA Design Tool (https://www.genscript.com/tools/gRNA-design-tool) and BLOCK-iT™ RNAi Designer, respectively. Both were synthesized by GenScript (Nanjing, China). The backbone of sgFAM83A used pSpCas9 BB-2A-Puro (Ran et al., 2013).Transient transfection of sgRNA-Cas9 plasmid and FAM83A overexpression plasmid were performed via Lipofectamine 3000 (Thermo Fisher Scientific, Waltham, MA, USA). The infected cells were subject to puromycin (1 μg/ml) selection for 5 days after transfection. Individual puromycin resistant colonies were picked up manually and then expanded in 12-well plates. Colonies carrying a deletion allele were checked by Western blots. Lipofectamine iMAX (Thermo Fisher Scientific, Waltham, MA, USA) was performed for siRNA transfection. FOXM1 cDNA, which was synthesized by Genscript (the synthetic sequences were referenced to Addgene #73237) and cloned into the pcDNA3.1 expression vector. According to the instructions, Lipofectamine 3000 (Thermo Fisher Scientific, Waltham, MA, USA) was transfected. Oligonucleotide sequences are listed in Table S1.

Experimental animals

The Institutional Animal Care and Use Committee of Xuzhou Medical University provided full approval for this research (IACUC-2202049). BALB/c nude mice were acquired from GemPharmatech, Inc. (Nanjing, China). BALB/c nude mice were obtained from GemPharmatech, Inc. (Nanjing, China). The participants were randomly divided into two groups (n = 5, with no blinding performed in either group). Cells were injected subcutaneously into the axilla of the nude mice, with each mouse receiving a dosage of 1 × 107 cells. The maximum diameter of the tumor was measured every 2 days after implantation. After 12 days, the mice were euthanized, and the tumor was removed and weighed. AB line zebrafish were acquired from China Zebrafish Resource Center (Wuhan, China). They were randomly divided into two groups (n = 16, no blinding was performed, respectively). After A549 cells incubated with CFSE (Thermo Fisher, Waltham, MA, USA), xenograft perivitelline injection were conducted in 2 day-post-fertilization zebrafish via a Picoliter Microinjector (PLI-100A; Warner Instruments, Hollister, MA, USA) with a glass capillary needle (Sutter, Q100-50-10) made on a laser-based needle puller (P-2000; Sutter, Sacramento, CA, USA). Observed cell extravasation in the caudal vasculature at 3 days post-transplant and imaging on a Zeiss AxioZoom V16 fluorescence microscope.

Dual-luciferase reporter assay

The FOXM1 promoter truncature plasmid were inserted into the pGL3 basic vector (Promega Corporation, Madison, WI, USA). All were co-transfected with a pRL-TK plasmid into cells by using Lipofectamine 3000 (#L3000075; Thermo Fisher Scientific, Waltham, MA, USA) in triplicate. Luciferase activity was measured using the Dual-Glo Luciferase Assay System (#DD1205-01; Vazyme, Nanjing, China) according to the manufacturer’s guidelines. Results represent the mean ± standard deviation (SD) of three experiments.

Immunohistochemistry (IHC)

The LUAD tissues were embedded in paraffin after being treated in 10% formalin. The tissues were then divided into 5 μm-thick sections, and the sections were treated with the primary antibodies anti-FAM83A (#20618-1-AP), anti-Ki-67 (#27309-1-AP), and anti-FOXM1 (#13147-1-AP) (Proteintech, China) for an overnight period. The sections were then stained with a 3,3-diaminobenzidine solution and treated with an HRP-polymer-conjugated secondary antibody (CST, San Antonio, TX, USA) for 1 h at 37 °C.

Immunofluorescence microscopy

Cells were permeabilized with PBS containing 0.05% Triton X-100 after being fixed with 4% paraformaldehyde for 30 min. Cells were washed three times with PBS before being incubated with FOXM1 primary antibody (#ab207298, dilution: 1/200; Abcam, Boston, MA, USA) for a whole night at room temperature. Following three PBS washes, the cells were treated for 1 h with a secondary antibody that was FITC-conjugated. The ZEISS confocal system took pictures of fluorescence (Lsm710, Zeiss, Baden-Württemberg, Germany).

Total RNA isolation, reverse transcription, and qRT PCR

Using the Total RNA Isolation Kit (TIANGEN, Sichuan, China), total RNA was isolated from issues, and cell lines and kept in a freezer at −80 °C. All complementary DNA (cDNAs) were created using the GoldenstarTM RT6 cDNA synthesis kit (TSINGKE, Sichuan, China) in accordance with the user’s manual and kept in a freezer at −80 °C. ABI 7,500 equipment (Applied Biosystems, Foster City, CA, USA) and ChamQ Universal SYBR qPCR Master Mix (Vazyme, Shuangbai Alley, China) were used to conduct qRT-PCR experiments. Table S1 displays the sequence of primers.

Western blot

The media was discarded, and RIPA lysate was added to the cells for total protein extraction. The total protein concentration was measured using the BCA technique (Thermo, Shanghai, China). Protein samples were separated through SDS-PAGE and transferred onto a PVDF membrane. After blocking with 5% skim milk at room temperature for 1 h, the membrane was washed three times with TBST and then incubated overnight at 4 °C with the primary antibody at the recommended dilution ratio. Subsequently, the membrane was washed again with TBST and incubated at room temperature for 1 h with the DyLight 800-labeled secondary antibody (ROCKLAND, USA). The membrane underwent three additional TBST washes before imaging using a fluorescence scanning instrument (Odyssey, Danbury, CT, USA). Primary antibodies against FOXM1 (ab207298, dilution: 1/1,000), FAM83A (ab128245, dilution: 1/1,000), CDC20 (ab183479, dilution: 1/2,000), CDC6 (ab109315, dilution: 1/4,000), cyclin B (ab32053, dilution: 1/8,000), H3 (ab1791, dilution: 1/5,000), and GAPDH (ab9485, dilution: 1/2,500) were bought from Abcam for this work.

Nucleus–cytoplasm fractionation

First, two washes of precooled PBS were performed on 1 * 106 LUAD cells. After scraping the cell layer into 500 l of PBS, it was centrifuged at 500 g for 5 min at 4 °C. The supernatant was then eliminated, and the washing procedure was repeated twice. In the end, LUAD cells in culture were used to isolate nuclear and cytoplasmic protein using NE-PER Nuclear and Cytoplasmic Extraction Reagents (Thermo Fisher Scientific, Waltham, MA, USA) in accordance with the manufacturer’s instructions.

Cell cycle analysis with flow cytometry

After trypsin digestion, cells were collected, twice-washed with PBS, and overnight fixed in ice-cold 70% ethanol. Fixed cells were rinsed twice with PBS before being stained for 30 min at 37 °C in PBS containing propidium iodide (PI, 50 g/mL) and RNase (100 g/mL). On a Beckman Coulter Epics XL-MCL flow cytometer, cell cycle analysis was carried out using System II (version 3.0) software (Beckman Coulter, Brea, CA, USA).

Transwell and matrigel invasion assay

Transwell assay was used to test the LUAD cells’ capacity to invade 24-well Boyden chambers with precoated Matrigel (Corning Incorporated, Corning, NY, USA). Briefly, 1× 105 cells in 200 μL of serum-free media were plated into the top chamber, and 600 μL of RPMI 1,640 medium with 10% FBS was added to the bottom chamber. Cells on the upper surfaces of the transwell chambers were removed after 24 h of incubation at 37 °C with 5% CO2, and cells that had crossed to the bottom surface were fixed with 4% paraformaldehyde and stained with 0.1% crystal violet. A fluorescence microscope (Zeiss, Baden-Württemberg, Germany) was used to collect the invading cells, which were then counted from five randomly selected areas.

5-Ethynyl-2′-deoxyuridine (EdU) assay

After being planted onto 96-well templates, A549 and PC9 cells were collected 48 h after transfection. After that, cells were treated with 50 mM EdU (#KGA331; KeyGEN, Nanjing, China) for 2 h, fixed with 4% paraformaldehyde, and labeled with Apollo Dye Solution. DAPI was used to counterstain the cell nuclei. A fluorescence microscope (Zeiss, Baden-Württemberg, Germany) was used to spot growing cells with green signals.

Colony formation assay

A total of 300 cells with FAM83A knockdown or NC were seeded in 6-well plates over the course of one night, treated with DMSO (NC), or palbociclib, and then cultivated for an additional 14 days. The cells were then fixed in 4% formaldehyde in PBS (vol/vol) for 10–15 min, washed twice with PBS to remove floating cells, and stained for 20 min in 0.1% crystal violet/10% ethanol. Colonies were observed using an Odyssey Imaging System (LI-COR) after the staining solution had been aspirated, three water washes, air drying, and visualization.

Cell counting kit-8 (CCK8)

Cells with FAM83A knockdown or NC incubated in 96-well plates were treated as indicated and cell proliferation was assessed by CCK-8 assay (SAB Biotech, College Park, MD, USA) at 0, 24, 48 and 72 h post treatment following the manufacturer’s instruction. Optical density (OD) was recorded at 450 nm.

Chromatin immunoprecipitation (ChIP) assay

ChIP was carried out using the Magna ChIP A kit (#17-610; MilliporeSigma, Burlington, MA, USA) in accordance with the manufacturer’s instructions. Anti-Flag antibodies(#14793S) were purchased from CST Inc. The qPCR analysis was carried out as previously mentioned.

Statistical analysis and in silico bioinformatics analysis

R (version 4.1.2) and GraphPad Prism 8 were used to conduct the statistical analysis. The difference between the several groups was compared, and the data are given as the means ± standard deviation (SD). Analysis of differentially expressed genes (DEGs) was performed using the “DESeq2” R package. GSEA assay were conducted via “clusterProfiler” R package. Copy number assay were performed by “maftool” R package. All published data used in this study reveal in Data Availability. The t test or one-way ANOVA were used to examine differences between the various groups. All assay was performed three independent times. At p < 0.05, a difference was deemed significant.

Results

FAM83A was screened as a histological subtype-related transcription factor in LUAD

H&E-stained whole slide images of 541 patients from the TCGA-LUAD database were obtained, and the histological subtypes and proportions of tumor tissues were evaluated, including nine patients with predominance of low-risk pathological subtypes (>50% of lepidic lung adenocarcinoma), 107 patients with predominance of intermediate-risk pathological subtypes (>50% of acinar and papillary lung adenocarcinoma), and 73 patients with predominance of high-risk pathological subtypes (>50% of micropapillary and solid lung adenocarcinoma). Transcription factors play a pivotal role in shaping the molecular evolution patterns and transcriptional profiles of malignant tumors (Goossens et al., 2017; Yu, Pardoll & Jove, 2009). Meanwhile, a variety of transcription factors has been identified as the key determinants of molecular subtypes in malignant tumors. To further explore the differences in transcription factor expression profiles among histological subtypes of lung adenocarcinoma, we analyzed the differences in transcription factor RNA expression profiles among three risk groups, which indicated that 26 transcription factors displayed specifically highly expression in high-risk pathological subtypes, whilst 24 and 23 transcription factors were specifically highly expressed in low-risk and intermediate-risk pathological subtypes, respectively (Figs. 1A, S1A and Table S2). Data retrieved from TCGA-LUAD database and GTEx database (Tang et al., 2017) revealed that transcription factor FAM83A displayed the most significant increase in expression levels within LUAD tissues (log2(FC) = 4.715) (Fig. 1B). Meanwhile, FAM83A was significantly increased in a variety of malignant tumors including LUAD, ESCA, LUSC, and UCEC, suggesting that it could serve as a malignant phenotypic transcription factor on a pan-cancer scale (Fig. 1C). Subsequently, survival analysis in the TCGA-LUAD database and multiple large-scale LUAD high-throughput cohorts (Bild et al., 2006; Okayama et al., 2012; Rousseaux et al., 2013; Xie et al., 2011) revealed that patients exhibiting high FAM83A expression had significantly poorer overall survival rates and progression-free rates (Figs. 1D–1G). Copy number analysis conducted on tumor tissues and adjacent tissues extracted from LUAD patients demonstrated significant copy number amplification at the FAM83A locus in lung adenocarcinoma tissues (Fig. 1H), which could potentially explain the observed high expression of FAM83A in tumor tissues. We continued to explore the distribution of FAM83A expression in patients with lung adenocarcinoma. It was observed that FAM83A exhibited significantly higher expression levels in patients diagnosed with advanced stage lung adenocarcinoma, as opposed to those diagnosed at earlier stages (Figs. S1B–S1D). We also detected the mRNA expression of FAM83A in 48 pairs of lung adenocarcinoma and adjacent tissues, which showed that FAM83A was significantly overexpressed in lung adenocarcinoma tissues (Fig. S1E), consistent with previous analysis results. Taken together, these data revealed that the expression of transcriptional factor FAM83A was specially increased in the high-risk group among histological subtypes, and also associated with poor outcomes and advanced stages in LUAD.

Figure 1 Characteristics of FAM83A in TCGA-LUAD.

(A) The heatmap revealed differentially expressed mRNA in low-risk pathological subtypes, intermediate-risk pathological subtypes, and high-risk pathological subtypes patients. (B) The mRNA expression of highly expressed transcription factors in high-risk patients was different between tumor tissues and adjacent tissues via TCGA and GTEx database. (C) Difference of FAM83A mRNA expression in cancer tissues and adjacent tissues at pan-cancer level in TCGA database. (D) Hazard ratio of FAM83A mRNA expression for tumorigenesis in multiple malignancies. (E) Survival analysis of LUAD patients with high and low FAM83A expression in TCGA database for overall survival. Survival analysis of LUAD patients with high and low FAM83A expression in previous cohort study for overall survival (E) and progression-free survival (F). (G) Copy number variation in the FAM83A chromatin region of LUAD and paracancinous tissues in TCGA database. **p < 0.01, ***p < 0.001 (Student’s t test).

FAM83A promotes LUAD tumor progression in vitro and in vivo

We evaluated FAM83A mRNA expression levels in multiple LUAD cell lines and normal human alveolar epithelial cells via qRT-PCR, which uncovered that FAM83A expression levels were higher in all six LUAD cell lines than in HBE cell line (Fig. 2A). For functional experiments, A549 cells featuring the lowest expression of FAM83A, and PC-9 cells exhibiting the highest expression of FAM83A were selected. We confirmed that the expression level of FAM83A was significantly decreased and increased when PC9 cells transfected with sgRNA and A549 cells transfected with overexpression, respectively (Fig. S1F). Subsequently, we performed colony formation and CCK8 assays in A549 and PC9 cells to evaluate the proliferative function of FAM83A, which showed that knockdown of FAM83A suppressed cell proliferation in both A549 and PC9 cells, while overexpression promoted cell growth in those cells (Figs. 2B, 2C, and S2A). Furtherly, we conducted EdU assays in A549 cells, and the knockdown of FAM83A inhibited A549 cell proliferation while the overexpression of FAM83A increase the proliferation (Figs. 2D and S2B). We also examined the effect of FAM83A on cell invasion and migration in normal Transwell or coating with Matrigel. As expected, FAM83A upregulation increased cell invasion and migration capability, whereas reduced FAM83A expression led to the opposite results both in A549 and PC9 cells (Figs. 2E, 2F, S2C and S2D). Collectively, these data suggest that FAM83A could promote the malignant phenotype of LUAD in vitro.

Figure 2 The function of FAM83A in vitro.

(A) FAM83A mRNA expression in several LUAD cell lines and lung epithelial cell. (B–D) Colony formation, CCK8-cell counter assay and EdU assays of A549 and PC9 cells with FAM83A knockdown or overexpression. (E and F) Transwell assay and Matrigel of A549 and PC9 cells with FAM83A knockdown or overexpression. *p < 0.05, **p < 0.01, ***p < 0.001 (Student’s t test).

Furthermore, zebrafish and BALB/c nude mice tumor xenotransplantation models were used to evaluate the effect of FAM83A on LUAD progression in vivo (Fig. 3A). We administered A549-sgNC and A549-sgFAM83A cells via perivitelline injection in the zebrafish 3 days’ post-injection, decreased expression of FAM83A resulted in a significant reduction in the number of metastasis cells in zebrafish tail (Fig. 3B). At 2 weeks after injection of A549-sgNC and A549-sgFAM83A cells into the axilla of nude mice, downregulation of FAM83A significantly inhibited LUAD tumor growth, and tumor volume was smaller than that of the control group at the end of the experiment (Figs. 3C and 3D). The application of immunohistochemistry (IHC) staining revealed a marked downregulation of FAM83A and Ki-67 expressions in tumor tissues of the sgFAM83A group as opposed to that of the control group (Fig. 3E). Together, these results indicate that FAM83A promotes LUAD tumor progression in vivo.

Figure 3 The function of FAM83A in vivo.

(A) Design of in vivo experiments. (B) Perivitelline injection were conducted in 2dpf zebrafish larvae and observed cell extravasation in the caudal vasculature at 2 days post-transplant. (C and D) The of subcutaneous xenograft tumors in nude mice. (E) IHC staining showed the protein expression of FAM83A adn Ki-67 in xenograft tumor tissues. All the results were shown as mean ± SD (n = 3), which were three separate experiments performed in triplicate. *p < 0.05, **p < 0.01, ***p < 0.001 (Student’s t test).

FAM83A regulates the key genes involved in cell cycle at transcriptional level

To explore the possible transcriptional regulation of FAM83A in LUAD, we analyzed the difference in bulk RNA-seq data between patients exhibiting high FAM83A expression (TOP15%) and low FAM83A expression (Bottom15%) in TCGA-LUAD database. A total of 98 genes with high expression and 50 genes with low expression in top 15% were screened out (Fig. 4A and Table S3). Pathway enrichment analysis showed that patients with high FAM83A expression were significantly enriched in cell cycle-related pathways (Fig. 4B). These results suggested that FAM83A may be involved in the transcriptional activation of cell cycle key genes. The above results were also validated in GSE30219 (Figs. 4C, 4D, and Table S4). Subsequently, we identified key genes that are known to play a role in cell cycle regulation from the selected pool of highly expressed genes within the top 15% (CDC6, CDC20, FOXM1, CDKN3), and assessed the extent of the correlation between FAM83A and these genes within PC-9 cells, which showed that FAM83A could significantly positively regulate the transcription of the aforementioned genes (Fig. 4E). We also verified that certain proteins were related to cell cycle pathway (FOXM1, cyclin B, CDC20 and CDC6), and obtained corresponding results (Fig. 4F). Moreover, the FACS results also indicated that reduced FAM83A expression increased cell G1/G0 cell arrest, whereas FAM83A upregulation led to the opposite results (Fig. 4G). These results confirmed that FAM83A could increase the key genes in transcriptional level to regulate the cell cycle.

Figure 4 The potential mechanism of FAM83A.

(A) The heatmap revealed differentially expressed mRNA between high FAM83A expression patients (top 15%) and low FAM83A expression patients (bottom 15%) from TCGA database. (B) The pathway enrichment for differentially expressed mRNA in TCGA database via GSEA. (C) The heatmap revealed differentially expressed mRNA between high FAM83A expression patients (top 15%) and low FAM83A expression patients (bottom 15%) from GSE30219. (D) The pathway enrichment for differentially expressed mRNA in TCGA database via GSEA. The mRNA level of cell cycle related genes in A549 cells after knockdown or overexpression of FAM83A was determined by qRT-PCR (E) and WesternBlots (F). (G) The cell cycle assay conducted in A549 cells after knockdown or overexpression of FAM83A. *p < 0.05, ***p < 0.001 (Student’s t test).

FAM83A hijacks the transcriptional regulation of FOXM1 to control the cell cycle

We proceeded to investigate the molecular mechanism by which FAM83A promoted transcription of key cell cycle genes. We firstly explored the correlation between the expression of FAM83A and the aforementioned cell cycle related genes, including FOXM1, CDC6, CDC20, and CDKN3, which revealed that FAM83A exhibited a more significant positive correlation with FOXM1 (P = 6.9e−05, R = 0.18) than others in TCGA-LUAD (Fig. 5A). FOXM1 is a forkhead box transcription factor that is involved in regulating cell proliferation, differentiation and tumorigenesis (Kalathil, John & Nair, 2021; Katzenellenbogen, Guillen & Katzenellenbogen, 2023; Liao et al., 2018). FOXM1 promotes tumor cell proliferation by activating genes involved in cell cycle progression and DNA synthesis. It drives cancer cells to enter and pass through the G1/S and G2/M cell cycle checkpoints (Alvarez-Fernández & Medema, 2013; Chen et al., 2013). To examine the ability of FAM83A as a transcription factor binding specific chromatin region, we designed a Flag tag fused plasmid containing FAM83A open reading frame (ORF) (Fig. 5B, upper), and designed four pairs of 500 bp primers at the promoter of FOXM1. ChIP-PCR results showed that Flag antibody significantly bound to the region 500 bp upstream of FOXM1 transcription start site (Fig. 5B, bottom). The truncated luciferase reporter plasmids with intervals of 500 bp according to the FOXM1 promoter were designed (Fig. 5C, left), which confirmed that the first 500 bp region upstream the transcription start site (TSS) was the core region of FOXM1 transcriptional regulation (Fig. 5C, right). FOXM1 is a transcription factor that serves as a crucial regulator of cell cycle progression, thereby controlling cell proliferation, DNA replication, and mitosis by modulating the expression of genes that are vital for these processes (Tan, Raychaudhuri & Costa, 2007). Nucleocytoplasmic separation analysis and immunofluorescence localization experiments revealed that increased FAM83A expression significantly enhanced the nuclear expression of FOXM1 (Figs. 5D and 5E). In conclusion, these results suggested that FAM83A hijacked transcription of FOXM1, the key gene of cell cycle, by binding to upstream 500 bp region of the promoter.

Figure 5 FAM83A hijacks the transcriptional regulation of FOXM1.

(A) The correlation of FAM83A and FOXM1, CDC6, CDC20 and CDKN3 in TCGA-LUAD database. (B) Upper: Schematic representation of the Flag and FAM83A fusion plasmid. Bottom: ChIP-PCR of FAM83A in promoter regions of FOXM1. (C) Left: Schematic representation of the FOXM1 promoter region truncated luciferase reporter plasmids. Right: Luciferase reporter gene assay for FOXM1 promoter region truncated plasmids. (D) Immunofluorescence assay for FOXM1 in A549 cells with overexpression FAM83A. (E) The expression of FOXM1 was detected after nuclear cytoplasmic separation assay in A549 cells with overexpression FAM83A. *p < 0.05, **p < 0.01, ***p < 0.001 (Student’s t test).

FAM83A regulated the cell cycle dependent on FOXM1

Furthermore, we further explored whether FAM83A promoted malignant progression in LUAD cells dependent on FOXM1. Colony formation and FACS cell cycle assays revealed that cell proliferation and cell cycle suppressed by FAM83A knockdown were reversed by co-transfection with FOXM1 overexpression plasmid. Conversely, the overexpression of FAM83A was found to promote LUAD cell proliferation and mitosis, while co-transfection with FOXM1 siRNA-pool resulted in a dampening effect on these outcome (Figs. 6A, 6B, S3A and S3B). Palbociclib is a selective inhibitor of the cyclin-dependent kinases CDK4 and CDK6 (Rocca et al., 2014). By inhibiting CDK4/6, Palbociclib ensures that the cyclin D-CDK4/6 complex cannot aid in phosphorylating Rb, which prevents the cell from G1 phase, and in turn from proceeding through the cell cycle (Xu et al., 2017). Therefore, we continued to explore whether Palbociclib could effectively restrict the malignant progression of cancer cells with high FAM83A expression. The Colony formation showed that Palbociclib could only slightly inhibit cell proliferation and cell cycle in LUAD cells A549 with low FAM83A expression. However, Palbociclib significantly inhibited cell proliferation and cell cycle in A549 cells overexpressing FAM83A (Figs. 6C, 6D, and S3C). This suggested that tumors exhibiting high FAM83A expression regulated cell cycle dependent on FOXM1, which may benefit from Palbociclib treatment.

Figure 6 FAM83A regulates cell cycle dependent on FOXM1.

Colony formation (A) and cell cycle assays (B) revealed that cell proliferation affected by FAM83A knockdown or overexpression was reversed by cotransfection with FOXM1 knockdown or overexpression. All the results were shown as mean ± SD (n = 3), which were three separate experiments performed in triplicate. **p < 0.01, ***p < 0.001 (Student’s t test). Colony formation (C) and cell cycle assays (D) revealed that the efficacy of Palbociclib was more significant after FAM83A overexpression.

FAM83A/FOXM1 axis positively correlates with malignant progression in LUAD patients

We measured the expression of FAM83A in 12 LUAD patients and divided them into two groups according to their respective FAM83A expression levels (Figs. 7A and 7B). The result demonstrated that patients presenting with high levels of FAM83A expression also displayed increased expressions of Ki-67 and FOXM1, as compared to patients exhibiting low FAM83A expression (Fig. 7C). Moreover, it was observed that patients with high levels of FAM83A/FOXM1 expression exhibited poorer patient outcomes, when compared to their low-expression counterparts (Fig. 7D). Additionally, the expression of CDK2 and CDC20, the keys genes involved in cycle cell, were found to be heightened in patients with high FAM83A/FOXM1 expression as opposed to those displaying low expression levels (Fig. 7E). Collectively, our data indicated that FAM83A/FOXM1 axis expression indicated malignant progression in LUAD patients.

Figure 7 FAM83A/FOXM1 axis correlates with poor prognosis in LUAD.

(A) Representative images of H&E staining, anti-ki67, anti-FAM83A and anti-FOXM1 immunohistochemistry in 12 LUAD patients. (B) According to the FAM83A IHC score of 12 LUAD patients, they were divided into high expression group and low expression group. (C) The IHC scores of Ki67 and FOXM1 were compared between the FAM83A high expression group and the FAM83A low expression group. (D) The overall survival analysis for patients with different expression of FAM83A and FOXM1 in TCGA-LUAD database. (E) The expression of cell cycle-related biomarker genes in patients with different expression of FAM83A and FOXM1 in TCGA-LUAD database. ***p < 0.001 (Student’s t test).

Discussion

Little is known about the biological function of transcription factors to determine the phenotype features of different histopathological subtypes among high-, intermediate-, and low-risk groups in lung adenocarcinoma. In our study, we identified nine transcription factors expressions associated with overall survival were specific increased in high-risk groups, among which FAM83A was most significantly increased in LUAD tissues compared with normal specimen, resulted from significant copy number amplification in tumor tissues. Patients with high FAM83A expression had worse overall and progression-free survival, and distributed in advanced stages. Gain- and loss-of-function assays revealed that FAM83A promoted cell proliferation, invasion, and migration of tumor cell lines both in vitro and in vivo. Mechanistic studies revealed that FAM83A, as the transcription factor, hijacked the promoter region upstream 500 bp of the FOXM1 TSS to regulate the transcriptional level of it. FAM83A regulated the cell cycle, thereby facilitating malignant progression via its regulation of FOXM1. Patients with active FAM83A/FOXM1 axis significantly score expressions of key genes in the cell cycle and poor prognosis in LUAD.

Previous reports have indicated that FAM83A possesses oncogenic properties in various types of cancer. Lee et al. (2012) revealed that FAM83A as a candidate cancer-associated gene conferred tumor cells with increased proliferation, invasion, and resistance to EGFR-tyrosine kinase inhibitors (EGFR-TKIs), which caused by phosphorylation of c-RAF and PI3K p85, the upstream of MAPK and downstream of EGFR. Marino et al. (2022) utilized the tissue microarray (TMA) of breast cancer to discovery a 1.5-fold increase in FAM83A expression in tumor tissues as compared to normal samples, which affected the expression of genes in cellular morphology and metabolism. Rong et al. (2020) revealed that the overexpressed FAM83A regulated the PI3K/AKT signaling pathway to mediate the development and progression of cervical cancer (CC). However, the biological function of FAM83A in cervical cancer is controversial. Xu & Lu (2020) discovered that although FAM83A expression was increased in cervical cancer compared with normal tissues, FAM83 knockdown increased the protein levels of α1, α3, α5, β4 and β5 integrins in vitro and in vivo, implicated the tumor-suppressive roles of FAM83A in cervical cancer. Additionally, FAM83A upregulated the expression of TSPAN1, thereby promoting autophagy in pancreatic cancer via the canonical WNT-CTTNB1 signaling pathway in pancreatic cancer (Zhou et al., 2021).

Some studies have provided evidence for a marked increase in the expression of FAM83A within lung cancer tissues (Gan, Li & Meng, 2020; Zhang, Sun & Mei, 2019; Zheng et al., 2020), leading to an increase in the expression of active β-catenin and Wnt target genes. However, existing knowledge regarding the expression patterns of FAM83A across histological subtypes in LUAD, and the mechanism underline the transcription factor, remains limited. In this study, our observations indicate that FAM83A expression is substantially elevated in the high-risk group as compared to those in the intermediate and low-risk groups, with a statistically significant correlation to both poor overall and progression-free survival in LUAD. Furthermore, patients displaying increased FAM83A expression were observed to have advanced stages. Functional assays revealed the oncogenic function of FAM83A, which promoted the malignant progression in LUAD. Additional observations of upregulated FAM83A expression indicated an added effect on the cell cycle, highlighting its dependence on the transcriptional regulation of FOXM1.

FOXM1 has critical roles in regulated tumorigenesis and malignant progression in lung cancer, including cancer therapy resistance, metastasis, and cell cycle progression. Previous study revealed that the yes-associated protein (YAP) and FOXM1 axis acted as a driver of epithelial-to-mesenchymal (EMT)-associated EGFR-TKI resistance, as well as the increased abundance of spindle assemble checkpoint (SAC) proteins (Nilsson et al., 2020). Patients with high FOXM1 expression were associated with a worse clinical outcome and correlated with expression of genes encoding SAC proteins, which revealed that YAP/FOXM1 axis along with SAC components are therapeutic vulnerabilities for targeting acquired resistance of EGFR-TKI. It also revealed that FOXM1 as a potential therapeutic target for cancer immunotherapy is associated with the modulation of PD-L1 expression (Madhi et al., 2022). The mechanistic study revealed that FOXM1 selectively upregulated PD-L1 expression via direct binding directly to the PD-L1 promoter. Additionally, FOXM1 had been found to be required for survival, quiescence and self-renewal of leukemia stem cells (LSCs), which directly binds to β-catenin to decrease its degradation, activating the Wnt/β-catenin signaling pathways (Sheng et al., 2020). Some studies have also revealed the mechanism underlying up-regulated FOXM1 expression in tumors. It has been reported that the Special AT-rich Binding Protein-2 (SATB2), which was one of several crucial nuclear matrix-associated proteins (NMPs), directly binds to the matrix attachment regions (MARs) sequence of the FOXM1 gene, thereby recruiting CBP to the MAR, which in turn activated FOXM1 expression, promoting the proliferation of glioma stem cells (GSCs) (Tao et al., 2020). However, little is known about the mechanism of increased FOXM1 expression in LUAD. In our study, we revealed that FOXM1 as the downstream target of FAM83A, which hijacked FOXM1 to directly bind the upstream 500 bp in TSS at the promoter region of FOXM1. We also observed the positive correlation between the expression of FOXM1 and FAM83A. FOXM1 acted as the key regulated gene involved in the progression of the cell cycle progression by directly binding to the promoter region of CCNE1, CCNB1, and CCND1, corresponding to cyclin E1, cyclin B1, cyclin D1 (Laoukili et al., 2005). However, it should be explored the downstream target of FOXM1 in LUAD to mediate the cell cycle progression upon the increased FAM83A expression pattern.

In our study, we use a pool of three siRNAs that achieves excellent short-term knockdown effects. Efficient suppression of the target gene expression can be maintained within 7 days after transfection of the siRNA pool. At the same time, siRNAs have the advantages of fast synthesis and convenient transfection. Therefore, in this study we used siRNAs against the potential downstream gene FOXM1 of FAM83A, rather than transfecting sgRNAs and performing single clone selection. However, we must acknowledge that siRNA has limitations in experiments that require long-term inhibition of expression, such as in vivo experiments.

Conclusions

In conclusion, our study identified that FAM83A was specifically increased in the high-risk group in LUAD when compared to the intermediate-, and low-risk groups. This increase in expression was correlated with poor overall and progression-free survival. Overexpression of FAM83A promoted the proliferation, invasion, and migration of tumor cells through the regulation of cell cycle progression. The mechanistic study revealed that FAM83A is directly bound to the upstream 500 bp of TSS in the promoter region of FOXM1 to mediate cell cycle regulation. Patients with active FAM83A/FXOM1 axis in LUAD had higher cell cycle abilities and a poor prognosis. Our study implicated that the FAM83A/FOXM1 is the therapeutic vulnerability for patients with higher expression of FAM83A.

Supplemental Information

Supplemental Information 1 Supplementary Figures and Tables.

Click here for additional data file.

Supplemental Information 2 Uncropped Gels/Blots.

Click here for additional data file.

Supplemental Information 3 Raw data.

Click here for additional data file.

Supplemental Information 4 Author Checklist.

Click here for additional data file.

We sincerely appreciate the China Zebrafish Resource Center, National Aquatic Biological Resource Center, NABRC-CZRC for providing the AB line zebrafish.

Additional Information and Declarations

Competing Interests

Author Contributions

Human Ethics

Animal Ethics

Data Availability

The authors declare that they have no competing interests.

Wei Fei conceived and designed the experiments, performed the experiments, analyzed the data, prepared figures and/or tables, authored or reviewed drafts of the article, and approved the final draft.

Yan Yan conceived and designed the experiments, performed the experiments, analyzed the data, prepared figures and/or tables, and approved the final draft.

Guangjun Liu conceived and designed the experiments, performed the experiments, analyzed the data, prepared figures and/or tables, and approved the final draft.

Bo Peng conceived and designed the experiments, performed the experiments, analyzed the data, prepared figures and/or tables, and approved the final draft.

Yuanyuan Liu conceived and designed the experiments, authored or reviewed drafts of the article, and approved the final draft.

Qiang Chen conceived and designed the experiments, authored or reviewed drafts of the article, and approved the final draft.

The following information was supplied relating to ethical approvals (i.e., approving body and any reference numbers):

The Ethics Committee of Jiangsu Cancer Hospital approved the study.

The following information was supplied relating to ethical approvals (i.e., approving body and any reference numbers):

The Institutional Animal Care and Use Committee of XMU provided full approval for this research (IACUC-2202049).

The following information was supplied regarding data availability:

The raw data is available in the Supplemental File.

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
