# Peer review of "High-risk histological subtype-related FAM83A hijacked FOXM1 transcriptional regulation to promote malignant progression in lung adenocarcinoma"

_PeerJ, doi:10.7717/peerj.16306_

## Round 0.1 · original submission · Major Revisions

Dear Dr. Chen,

Thank you for submitting your manuscript "High-risk histological subtype-related FAM83A hijacked FOXM1 transcriptional regulation to promote malignant progression in lung adenocarcinoma" to PeerJ. We have now received reports from the reviewers, and, after careful consideration, we have decided to invite a major revision of the manuscript.

As you will see from the reports copied below, the reviewers raise important concerns. We find that these concerns limit the strength of the study, and therefore we ask you to address them with additional work. Without substantial revisions, we will be unlikely to send the paper back for review.

If you feel that you are able to comprehensively address the reviewers’ concerns, please provide a point-by-point response to these comments along with your revision. Please show all changes in the manuscript text file with track changes or color highlighting. If you are unable to address specific reviewer requests or find any points invalid, please explain why in the point-by-point response.

Thanks

Abhishek Tyagi, PhD
Academic Editor,
PeerJ

Reviewer 1 ·

Basic reporting

The manuscript is well-written overall, with a professional tone and appropriate background information. The introduction and contextual grounding have no significant shortcomings. However, there are some aspects of basic reporting that could benefit from improvement. Below are some specific comments.

- Line 74: "Ryo and colleagues", line 78: "Thinh and colleagues", line 401 "Natascia and colleagues" were not properly cited. Please use the authors' last names and refer to PeerJ's reference format guide.
- In regard to figure captions, I recommend focusing solely on the description of the presented data, avoiding interpretations or conclusions. Using Figure 1B as an example, the current caption seems to be more appropriate for the results section of the main text as it includes interpretation. A more suitable caption might read, 'mRNA expression data of various transcription factors in tumor tissues compared to adjacent non-tumorous tissues, sourced from the TCGA and GTEx databases.' This approach would maintain the essential information yet adhere to the conventional structure of figure captions.
- Also, for all figure captions, please describe the statistical test conducted and the significance level definitions of the * annotation. Also, it is suggested to check and make sure to include the model cell line used to generate the data.
- Figure 1A: the image was not clear. The text labels were not legible. Please consider optimizing.
- Figure 1B: it is unclear whether any statistical tests were performed for transcription factors other than FAM83A. If such tests were done, it would be beneficial to report if any showed significant differences. This would give the reader a clearer understanding of the broader context in which FAM83A is being examined. Additionally, the basis for focusing specifically on FAM83A is not readily apparent. It is recommended to include more rationale for this choice.
- Supplemental Table S2: the presented mean expression data is not sufficiently informative. I recommend plotting the table into separate figure panels, like Figure 1B, in the supplemental figures. This way, we can visualize the spread of the data. I also recommend doing a statistical test for significance.
- Figure 1D-G: the figure captions appear mislabeled. Please verify and correct as needed.
- Figure 1H, the figure was not clear. Please improve the legibility of the axis numbering.
- Figure 2A – the caption and the main text do not fully describe the origins of the data or what experiments were conducted to obtain these results. It appears to be from a qPCR experiment. Please clarify.
- Figure 2B-D: the figure caption does not clearly describe what experiments were done in each of the panels. Please describe one at a time. Also, please show the necessary information on how the experiments were done (i.e., treatment times, number of replicates).
- Figure 2C: please indicate the number of biological replicates performed and what the error bars represent.
- Line 295: "We also examined cell invasion and migration capability of FAM83A". I suggest refining the phrasing to 'We also examined the effect of FAM83A on cell invasion and migration.' Your experiments are observing the potential effects of FAM83A on these processes rather than implying a direct involvement of FAM83A, which is not definitively established by these experiments.
- Lines 292-299: the authors discuss "cell proliferation" and invasiveness broadly. However, specific experiments were conducted using LUAD cell lines. The conclusions drawn from this experiment should be specified and contextualized within the realm of LUAD cell lines. This specificity is essential to avoid overgeneralization and ensure that the interpretation of the results is accurately confined to the observed context.
- Figure 5A: labels do not appear legible. Please optimize.
- A brief introduction to FOXM1 is needed earlier in the manuscript, instead of near the end of the results, at line 344.
- Supplemental Figure S3: in the Figure caption, please indicate the cell line model used.
- Line 398: "Sun-Young and colleagues" needs proper in-text citation.
- There are a lot of works cited and discussed in the Discussion section. It is recommended to review the relevance of each discussion to keep it concise and on topic.

Experimental design

The experimental methodologies employed in this study are, for the most part, robust and appropriate. However, there are some aspects that could benefit from further clarification or revision to enhance the rigor and reproducibility of the findings. Some specific comments and suggestions are provided below.

- Figure 6A seems to depict the effects of siRNA knockdowns on colony formation – which is usually a long-duration assay. It's important to note that siRNA effects are transient and tend to diminish over time. This makes siRNA less ideal for experiments over extended durations, as the effects of knockdown may not persist throughout the entirety of the assay. Similarly, if FOXM1 overexpression was achieved through transient means, the enhanced expression might not be maintained for the full duration of the experiment. I suggest that these considerations be factored into the interpretation of these results and possibly mentioned as a limitation in the discussion. Furthermore, employing more durable methods of gene expression manipulation, such as shRNA for knockdown or viral transduction for overexpression, might provide more consistent results over the length of the experiment.
- I also recommend trying FOXM1 knockdown in combination with FAM83A overexpression in experiments described in Figure 6, to understand if the effect of siFOXM1 could be fully compensated by FAM83A overexpression.
- Some experiments were not described: colony formation assay, FAM83A overexpression, and CCK8.
- The source of the overexpression plasmid for FAM83A was not provided.

Validity of the findings

The conclusions, for the most part, are substantiated by the experimental data. However, there are some areas that could be refined to enhance the validity and impact of the findings. Detailed comments on these potential improvements are provided below.

- Line 322-324: "… assessed the extent FAM83A's transcriptional regulatory capacity on these genes within PC-9 cells, which showed that FAM83A could significantly positively regulate the transcription of the aforementioned genes". The data presented demonstrate correlation but do not provide direct evidence of causation. While FAM83A may be associated with the transcription of these genes, without further functional or mechanistic studies yet, stating it as a definitive regulatory factor at the current point of the manuscript is premature and could potentially be misleading. I recommend sticking to the observation that there is an association while there are further validations in Figure 5.
- Line 346-348 "Nucleocytoplasmic separation analysis and immunofluorescence localization experiments revealed that increased FAM83A expression significantly enhanced the nuclear expression of FOXM1 (Fig. 5D-E)." In the presented immunofluorescence images, it appears that the overall signal intensity for FAM83A is noticeably lower than that of the mock control. I suggest the authors confirm whether the exposure settings remained consistent across all samples. Additionally, from the images, there seems to be an alteration in the cellular localization of FOXM1, a shift from the cytoplasm to the nucleus? This is a potentially interesting observation that may warrant further investigation. As for the Western blot in Figure 5E, it didn't convincingly demonstrate the claimed 'enhanced nuclear expression of FOXM1.' To provide more concrete evidence, I recommend quantifying the immunofluorescence images with more sample sizes, which would aid in validating these findings and giving a clearer picture of any changes in FOXM1 localization or expression levels.

Reviewer 2 ·

Basic reporting

The authors described a correlation of FAM83A level in LUAD with malignancy, patient survival and poor prognosis. They further demonstrated that in vitro FAM83A level correlates with tumor cell growth and tumorigenesis in zebrafish and nude mice. These findings are of interest to the field. In an effort to determine the molecular mechanism of how FAM83A contributes to tumor cell growth, the authors reports that FAM83A may regulate cell cycle through FOXM1. However, some results in this aspect are confusing to me.
1. In figure 2D, it is shown that FAM83A overexpression leads to more cells incorporating EdU; however, in figure 4G, there is no difference in the proportion of cells in S phase.
2. Why is there an increase in the proportion of cells in G2 when FAM83A is overexpressed?
3. Based on the results that FAM83A knockout causes an increase in G1, it is reasonable to speculate that it may play a role in G1/S transition. Therefore, it will be useful to look at factors involved in that process such as Rb.
4. Palbociclib treatment should block cells to G1, but the authors instead observed a decrease of G1 cells in 6D. What is the explanation here?

Other major points:
1. Is the effect of FAM83A observed in tumor cells specifically? Parallel molecular genetic experiments should be done in the control HBE cells.
2. The results from the Luciferease assay in 5C does not support the conclusion that FAM83A binds to -500 to TSS of FOXM1.

Minor points:
1. Figure 2D, 5D need quantification
2. The level of FOXM1 overexpression and knockdown needs to be demonstrated by WB

Experimental design

N.A.

Validity of the findings

N.A.

Additional comments

N.A.

Reviewer 3 ·

Basic reporting

The language used is clear and unambiguous for the most part. The references were cited appropriately and there was enough background. However, I would have liked to see more background in the intro about what is known about FAM83A in LUAD and how your study adds to the knowledge.

Experimental design

The research question was well defined Methods section can be improved at several places. For example in the confocal microscopy section, what is main antibody? Please mention the antibodies with Cat. no. and the ratios used. Similarly please provide the antibody ratios used in western blotting. Please provide more information on cloning of FOXM1 siRNA and FAM83 sgRNA with vector information. Which cells were infected in BalBc mice? How was tumor volume measured after implantation? Not enough information is provided for IHC. Which primary antibodies? Stains are not mentioned properly. Line 222 please explain how the fluorescence microscope was used to collect the invading cells.

Validity of the findings

The authors do a thorough job of to provide data that is robust and statistically sound.

In fig 2C, Why the FAM83A overexpression and downregulation shown in different cell lines. Can you please include the corresponding downregulation and overexpression results in supplementary. Figure 2D, which cell line is used? Please be mindful of this throughout the manuscript.
Although the correlation is a little weak, but I would still liked to have seen Luciferase activity between FAM83A and CDC6 and CDC 20 to rule it out as a potential binding partner. Moreover, Luciferase assay data does not corroborate with CHIP-qPCR data that FAM83A binds 500 bp upstream of TSS. Please explain. The title of result section 5 is very confusing. Results with Palbociclib, although lacking in vivo validation, looks promising in vitro. Maybe include those in the title. Also, which cell lines were used in figure 5?

---

## Round 0.2 · Minor Revisions

Dear Dr. Chen,
Thank you for your submission to PeerJ.
It is my opinion as the Academic Editor for your article - High-risk histological subtype-related FAM83A hijacked FOXM1 transcriptional regulation to promote malignant progression in lung adenocarcinoma - that it requires a number of Minor Revisions.

With kind regards,
Abhishek Tyagi
Academic Editor
PeerJ Life & Environment

Reviewer 1 ·

Basic reporting

The manuscript has significantly improved, and the authors have addressed most of the concerns from the previous round of review.
Below are a few additional comments:
- The figure captions were revised in the main text but not on the Figure captions uploaded with the figures.
- Figure 1A is still not very clear, and the gene names are not legible. This may perhaps be the result of the size of the panel. It is recommended to re-organize the figure panels to make sure that all information is legible when zoomed out.
- Line 269 “FAM83A was screening” should be “FAM83A was screened”?

Experimental design

The authors have adequately addressed the comments from the previous round of review.

Validity of the findings

The authors have adequately addressed the comments from the previous round of review.

Reviewer 2 ·

Basic reporting

I believe all my previous concerns have been successfully addressed by the authors. Therefore, I recommend publication of this study.

Experimental design

N.A.

Validity of the findings

N.A.

Additional comments

N.A.

---

## Round 0.3 · accepted · Accept

Dear Dr. Chen,

Thank you for your submission to PeerJ. Your manuscript has been Accepted for publication. Congratulations.


With kind regards,
Abhishek Tyagi


Reviewer 1 ·

Basic reporting

The authors have addressed all comments from the previous round of review.

Experimental design

no comment

Validity of the findings

no comment